# Polatuzumab vedotin combined with bendamustine and rituximab for relapsed/ refractory diffuse large B-cell lymphoma: A systematic review protocol

**Mohammadreza Eslami** [1,2], **Mahdi Mehrabi**[1,2], **Mehrdad Payandeh**[3], **Fakhredin Saba**[2]*

**1** Student Research Committee, Kermanshah University of Medical Sciences, Kermanshah, Iran, **2** School of paramedical, Kermanshah University of Medical Sciences, Kermanshah, Iran, **3** Department of Internal medicine, School of Medicine, Kermanshah University of Medical Sciences, Kermanshah, Iran

* saba.fakhredin@gmail.com

## Abstract

### Background

Diffuse large B-cell lymphoma (DLBCL) is an aggressive non-Hodgkin lymphoma subtype with a significant relapse rate and poor prognosis in relapsed/refractory (R/R) patients. Polatuzumab vedotin in combination with bendamustine and rituximab (Pola-BR) has demonstrated promising efficacy and safety as salvage therapy for R/R DLBCL. This systematic review protocol aims to comprehensively evaluate the efficacy of Pola-BR for the treatment of R/R DLBCL by synthesizing data from relevant randomized controlled trials.

### Methods

This protocol details the eligibility criteria, search strategy, study selection, data extraction, and analysis methods for the systematic review. Randomized controlled trials comparing Pola-BR with other interventions for R/R DLBCL will be included. The primary endpoint is overall survival, with secondary endpoints being progression-free survival and incidence of adverse events. A comprehensive search will be conducted across databases such as Medline/PubMed, Cochrane Library, Web of Science, Scopus, EMBASE, ProQuest, EU Clinical Trials Register, WHO International Clinical Trials Registry Platform (ICTRP), and ClinicalTrials.gov from the January 2000 to April 2024. To assess the potential risk of bias, the Cochrane Risk of Bias 1 tool will be used. Data synthesis will utilize fixed-effect or random-effects models, and subgroup and meta-regression analyses will examine heterogeneity. Additionally, publication bias and sensitivity analyses will be performed, and the GRADE approach will be applied to assess the certainty of the evidence.

### Conclusion

This systematic review and meta-analysis protocol provides a rigorous framework for evaluating the efficacy of Pola-BR in the treatment of R/R DLBCL. The results will inform clinical decision-making and guideline development, addressing the unmet need for effective and

**Data Availability Statement:** No datasets were generated or analysed during the current study. All

relevant data from this study will be made available upon study completion.

**Funding:** The author(s) received no specific funding for this work.

**Competing interests:** The authors have declared that no competing interests exist.

tolerable treatments for this challenging patient population. Potential limitations and biases will be acknowledged, and future research directions will be discussed.

## Background

Diffuse large B-cell lymphoma (DLBCL) represents the predominant subtype of non-Hodgkin lymphoma, and accounts for 20.0% to 63.9%of all diagnosed cases. The global incidence of DLBCL varies considerably, ranging from 2.3 to 13.8 cases per 100,000 person-years [1]. DLBCL is an aggressive and heterogeneous disease with variable clinical outcomes and responses to treatment. The conventional first-line therapy for DLBCL is combination of rituximab with cyclophosphamide, doxorubicin, vincristine, and prednisone (R-CHOP). This treatment protocol results in a complete response (CR) rate of approximately 60% and a 5-year overall survival (OS) rate of about 70% [2]. However, up to 25% of patients are either refractory to R-CHOP or relapse after an initial response [2]. These patients have a poor prognosis, with a median OS of less than one year [2]. Therefore, there is an unmet need for effective and tolerable salvage therapies for relapsed or refractory (R/R) DLBCL.

Multiple treatment modalities have been developed for R/R DLBCL, each with varying degrees of effectiveness. Salvage chemotherapy such as R-ICE (rituximab, ifosfamide, carboplatin, etoposide) or R-DHAP (rituximab, dexamethasone, high-dose cytarabine, cisplatin) are commonly used. However, outcomes with salvage chemotherapy alone are suboptimal, with 5-year survival rates only 21% [3].

High-dose chemotherapy followed by autologous stem cell transplantation (HDT-ASCT) remains the standard of care for eligible patients with chemosensitive R/R DLBCL. The landmark PARMA trial established HDT-ASCT as a standard approach and demonstrated recovery rates of 30–40% in patients who relapsed after initial therapy. However, outcomes are significantly worse in patients with primary refractory disease or early relapse (within 12 months of initial therapy) [4].

For patients ineligible for HDT-ASCT or for whom this approach has failed, chimeric antigen receptor (CAR) T cell therapies targeting CD19 have shown promise. Axicabtagene ciloleucel achieved an overall response rate of 82% in the ZUMA-1 trial, with 54% of patients achieving a complete response [5]. Similarly, lisocabtagene maraleucel demonstrated a complete response rate of 66% in the TRANSFORM trial for second-line treatment of R/R DLBCL [6].

Despite these advances, there remains a significant unmet need for effective and tolerable salvage therapies for R/R DLBCL, particularly for patients who are treatment ineligible or who have failed the above treatments.

Polatuzumab vedotin (Pola) is an innovative antibody-drug conjugate that targets CD79b, a part of the B-cell receptor complex commonly found in various B-cell malignancies. Pola delivers the potent cytotoxic monomethyl auristatin E to CD79b-positive cells, triggering cell cycle arrest and apoptosis [7]. Bendamustine (B) is a bifunctional alkylating agent with cytotoxic and immunomodulatory effects [8]. Rituximab (R) is a monoclonal antibody that binds to CD20, a surface antigen expressed on most B cells, and mediates antibody-dependent cellular cytotoxicity, complement-dependent cytotoxicity, and apoptosis [9].

The combination of Pola, B, and R (Pola-BR) has emerged as a promising therapeutic option for R/R DLBCL. In the GO29365 study, Pola-BR demonstrated superior efficacy compared to BR alone, achieving a complete response rate of 42.5% and median overall survival of 12.4 months, versus 17.5% and 4.5 months, respectively [10]. These results suggest a potential advantage over historical data for conventional salvage therapies, warranting further investigation.

The primary outcome measure of this systematic review is OS, which is defined as the duration from the time of randomization or the initiation of treatment to death from any cause [11]. OS is considered the most clinically relevant endpoint in oncology trials as it reflects the direct impact of the intervention on patient survival. This study will evaluate secondary outcomes, including both progression-free survival (PFS) and the incidence of adverse events (AEs). PFS is typically defined as the time from randomization or initiation of the treatment to disease progression (based on radiological or clinical assessment) or death, whichever occurs first [11]. PFS is a commonly used surrogate endpoint in oncology trials, because it provides an earlier indication of treatment effectiveness than OS, although it cannot always reliably predict OS. AEs refer to all adverse medical events, regardless of their causal relationship to treatment, and are important for evaluating the safety and tolerability of the intervention [12].

Several systematic reviews have been conducted to evaluate the efficacy and safety of salvage therapies for R/R DLBCL [13–16]. However, language limitations and the lack of a comprehensive search limited the scope of these studies. Therefore, it is important to conduct this systematic review to provide a comprehensive and up-to-date analysis of the available data on Pola-BR for R/R DLBCL and to compare its performance with other salvage therapies. This systematic review will inform clinical decision-making and guideline development for this challenging patient population.

## Protocol and registration

This systematic review protocol is registered in the PROSPERO database (registration number: CRD42024521126). The Preferred Reporting Items for Systematic Reviews and Meta-Analysis Protocols (PRISMA-P) are followed in the design of this systematic review protocol [17]. The results of the systematic review will be reported in accordance with the Preferred Reporting Items for Systematic Review and Meta-Analysis (PRISMA 2020 statement) [18].

## Methods

### The eligibility criteria

**Types of studies.** We will include randomized clinical trials (RCTs) with a concurrent control group and compare Pola-BR with other interventions for the treatment of R/R DLBCL. We will exclude non-randomized trials, single group before-and-after trials, case series, and case reports. We will also exclude cohort studies because they are not suitable for assessing the effectiveness of Pola-BR. In addition, we will exclude non-original studies, such as narrative reviews, systematic reviews, and scoping reviews.

**Types of participants.** We will enroll adult patients (18 years and older) who have a confirmed diagnosis of R/R DLBCL, regardless of gender, race, ethnicity, disease severity, or stage. We will exclude patients with other types of lymphoma, such as follicular lymphoma, mantle cell lymphoma, or Hodgkin lymphoma, or contraindications to Pola-BR or any of the comparator interventions.

**Interventions.** This review will include studies evaluating the experimental intervention of Pola-BR for the treatment of R/R DLBCL. Polatuzumab vedotin acts as an antibody-drug conjugate against CD79b, while Bendamustine is a chemotherapy agent, and rituximab is a monoclonal antibody directed against CD20. Eligible studies will include any dosage, duration, frequency, or route of administration of Pola-BR consistent with the manufacturer's recommendations or accepted clinical practice guidelines.

**Comparators.** Comparative interventions will be considered as follows: no intervention or placebo, standard treatments such as rituximab monotherapy, rituximab in combination

with chemotherapy, or stem cell transplantation, and other experimental interventions that include novel drugs, immunotherapies, or gene therapies.

**Outcomes of interest.**   Studies will be included if they reported at least one of the following outcomes:

*Primary outcome.*

- **OS:** It will measure the length of time from randomization (when participants are assigned to a treatment group) until death from any cause.

*Secondary outcomes.*

- **PFS:** This outcome will assess the time from randomization to disease progression (worsening), relapse (return of the disease), or death from any cause. PFS provides an indication of how long treatment delays disease progression.

- **AEs:** This category will record any adverse or unintended effects that participants experience as a result of treatment. The severity of these events will be assessed using the Common Terminology Criteria for Adverse Events, a standardized system for classifying adverse events in clinical trials.

Studies that do not assess the primary endpoint or explicitly aim to assess a prespecified endpoint will be excluded from this review.

### Search strategy

A comprehensive and systematic search will be conducted to identify studies evaluating the effectiveness of Pola-BR in the treatment of R/R DLBCL. The search strategy will employ a combination of controlled vocabulary terms and free-text keywords to capture relevant concepts. Key concepts include polatuzumab vedotin, rituximab, bendamustine, and diffuse large B-cell lymphoma. Controlled vocabulary terms from systems such as Medical Subject Headings (MeSH) and EMTREE will be used, supplemented by free-text keywords to account for synonyms. Boolean operators (AND, OR, NOT) and search tags (title, abstract, etc.) as well as truncations and parentheses will be used to ensure precise results. No language or publication status restrictions will be applied; studies in languages other than English will be translated and included if eligible. Both published studies and gray literature will be taken into account.

**Database search.**   The following electronic databases will be searched from inception to April 2024: Medline/PubMed, Cochrane Library, Web of Science, Scopus, EMBASE, ProQuest, EU Clinical Trials Register, WHO International Clinical Trials Registry Platform, and ClinicalTrials.gov. Search strategies are tailored to the specific syntax and functionality of each database.

**Gray literature search.**   The database search will be supplemented by gray literature searches, including unpublished or non-indexed sources. Relevant sources include:

1. Research reports from organizations such as NICE, ASCO, and ESMO.

2. Conference proceedings and abstracts from oncology meetings (e.g., ASH, EHA, ICML, ISPOR, SMDM, HTA).

3. Theses and dissertations from repositories such as ProQuest Dissertations.

4. Reference lists of eligible studies and relevant systematic reviews.

5. Manual search of three prominent hematological malignancy journals: Cancers, British Journal of Hematology, and Leukemia and Lymphoma, to ensure comprehensive literature

**Table 1. PubMed search strategy using population and intervention elements of the PICO framework, January 2000 to April 2024.**

| Population | Intervention | Time interval |
|---|---|---|
| ((Lymphoma* AND ("Large Cell*" OR "Large Lymphoid*" OR "Large B-Cell*" OR "Histiocytic*" OR "Large B cell*")) OR DLBCL) | ("ACD79B VCMMAE" OR "dcds 4501a" OR "fcu 2711" OR "monoclonal antibody DCDS4501A" OR "polatuzumab vedotin*" OR "polivy" OR "rg 7596" OR "ro 5541077*" OR "ro5541077*") AND ("4 [5 [bis (2 chloroethyl) amino] 1 methyl 1h benzimidazol 2 yl] butanoic acid" OR "bendamustin*" OR "belrapzo" OR "bendeka" OR "cimet 3393" OR "syb l 0501" OR "treanda") AND ("abp 798" OR "bcd 020" OR "bi 695500" OR "cmab 304" OR "ct p10" OR "gp 2013" OR "hlx 01" OR "ibi 301" OR "idec c2b8" OR "jhl 1101" OR "mabthera" OR "rituximab*") | 2000/01/01:2024/04/30 [dp] |

coverage. Experts in the field will also be contacted to identify additional or ongoing studies that are not publicly available.

**Search syntax.** The detailed search strategy for PubMed is provided in Table 1.

**Search documentation and management.** The search process will be documented and reported using the PRISMA flow chart. Details recorded will include the date, database, search terms, and number of records retrieved for each search, as well as any changes or updates to the strategy. Reference management software like Mendeley will be used to store, organize, and deduplicate records, which will be exported to screening software like Rayyan to facilitate the screening and selection process. A Comprehensive search will be carried out across all databases by the end of April 2024.

For the Web of Science and the Cochrane Library, we will use the integrated date filters in the databases themselves.

## Screening and selection

We will follow a two-step process to screen and select studies for our systematic review. In the first phase, two reviewers will independently review the titles and abstracts of the records retrieved from the search strategy against a predefined checklist based on the eligibility criteria. Reviewers will mark each record as included (+), excluded (-), or unclear (*) using the checklist. Records marked as included or unclear by either reviewer will be retrieved for full-text screening. Records marked as excluded by both reviewers will be discarded.

In the second phase, two reviewers will independently assess the full-text articles of the selected records for eligibility using the same checklist. The reviewers will confirm the inclusion or exclusion of each article based on the full-text information. Articles that meet all inclusion criteria and none of the exclusion criteria will be included in the systematic review. Articles that do not meet the inclusion criteria or meet any of the exclusion criteria will be excluded from the systematic review. The reasons for exclusion will be recorded and reported.

Any disagreements between reviewers in the screening or selection phase will be resolved through discussion and consensus. If consensus cannot be reached, a third reviewer will be brought in to make the final decision.

## Risk of bias assessment

When assessing the risk of bias for this systematic review, the Cochrane Risk of Bias 1 (ROB-1) tool [19] will be used to assess included randomized controlled trials. Despite improvements

to ROB-2 intended to address ROB-1's limitations, several studies provide compelling reasons for the continued preference for ROB-1 in certain contexts. Notably, ROB-2 struggles with low interrater reliability (IRR), reporting an overall IRR of $\kappa = 0.16$ and moderate agreement limited to the "randomization process" domain [20]. This inconsistency highlights the practical difficulties inherent in ROB-2's detailed framework that can make consistent assessments across different raters difficult. In contrast, ROB-1's simpler structure may promote more consistent assessments, thereby increasing the reliability of systematic reviews.

Furthermore, the resource-intensive nature of ROB-2 represents a significant drawback, with systematic reviews indicating an average time commitment of approximately 358 minutes per study [21]. This considerable time investment can be prohibitive, especially in large-scale reviews or resource-constrained settings. The complexity of ROB-2, which includes new terminology, detailed signaling questions, and intricate domain-specific approaches, demands extensive training and calibration, that may not always be feasible. Additionally, adherence to ROB-2 guidance has been frequently poor. A meta-epidemiological study found that only 69.3% of systematic reviews met required application at the outcome measurement level. For assessments with multiple primary endpoints, this adherence drops to 28.8%. The lack of adherence was particularly evident in low-quality reviews, where ROB-2 was applied at the study level rather than the outcome measure level [22]. These factors contribute to the frequent continued use of ROB-1, which remains more prevalent in non-Cochrane systematic reviews due to its user-friendly design and reduced resource demands [23]. Additionally, ROB-1's straightforward approach has proven advantageous in specific contexts, such as acupuncture trials, where it resulted in a higher rate of high-risk assessments compared to ROB-2 [24]. Consequently, while ROB-2 offers a more nuanced assessment, the practical benefits of ROB-1 —particularly in ensuring consistency, feasibility, and adequate application in systematic reviews—support its sustained relevance in bias assessment.

Although the drawbacks of ROB-1, such as subjective interpretations and lack of nuance in certain areas, were acknowledged, three critical areas of potential bias were selected. These areas were chosen as the most relevant to the methodology of the studies under investigation: random sequence generation (selection bias), blinding of outcome assessment (detection bias), and handling of incomplete outcome data (attrition bias). An in-depth scientific analysis will assess the risk of each area across studies.

The overall risk of bias judgment will be derived from the responses to these three pivotal ROB-1 domains. The studies that answered "yes" to all three questions will be classified as low risk, those that answered "yes" to two questions will be classified as moderate risk, and those that answered "yes" to zero or one "answered will be considered to be at high risk of bias. However, all ROB-1 domains are subjected to careful assessment, with cumulative results presented visually via a traffic light plot indicating different levels of risk of bias.

Two independent reviewers carry out a risk assessment for each study included in the analysis. If there are discrepancies between the reviewers' assessments, an attempt will first be made to find a solution through consultation and discussion. If consensus cannot be reached after consultation, a third reviewer will be consulted who will make the final decision on the risk of bias.

## Data extraction

We will extract the following data from each eligible study using a standardized data extraction form:

- General information: study ID, authors, year of publication, journal, country, funding source, conflict of interest

- Study characteristics: study design, sample size, inclusion and exclusion criteria, randomization method, allocation concealment, blinding, duration of follow-up, loss to follow-up, intention-to-treat analysis

- Intervention characteristics: name, dose, frequency, duration, route of administration, comparator, cointerventions, and adherence

- Outcome characteristics: primary and secondary outcomes, definition, measurement, time points, unit of analysis, statistical methods, effect estimates, confidence intervals, p-values, subgroup analyses, sensitivity analyses, adverse events

Two review authors will independently extract data from each study using paper forms. We will use Stata software to code and categorize the extracted data. If necessary, we will compare the data extraction forms and resolve any discrepancies through discussion or consultation with a third review expert.

The corresponding authors of the included studies will be contacted to obtain missing data or to clarify any ambiguities in the reported data. We will make up to three attempts to contact the authors at different times. We will exclude the study from analysis if we do not receive a response or the required data.

If the data are presented in graphical format, we will use the online tool WebPlotDigitizer to extract the numerical data from the graphs.

## Meta-analysis

**Objective.**   This meta-analysis aims to synthesize existing evidence on the efficacy and safety of combination therapy with Pola-BR for the treatment of DLBCL. The analysis will be carried out on the condition that at least five studies are included to ensure a reliable statistical analysis [25].

**Statistical methods.**   The synthesis will use either a fixed-effects or a random-effects model based on the degree of methodological heterogeneity observed in the included studies. For the fixed effects model, the inverse variance method will be used, while for the random effects model, the Der Simonian and Laird method will be used.

**Effect measures.**

- **Dichotomous Outcomes:** Risk ratios will serve as the effect measure.

- **Continuous Outcomes:** Mean Differences will be used for outcomes measured on identical scales, while Standardized Mean Differences will be used for outcomes measured on different scales.

- **Primary and other time-to-event outcomes:** These outcomes will be analyzed using hazard ratios as an effect measure for time-to-event data.

**Heterogeneity assessment.**   Heterogeneity between the studies will be quantified using the $I^2$ statistic with the following thresholds:

- **Low:** 0–25%

- **Moderate:** 25–50%

- **Substantial:** 50–75%

- **High:** 75–100%

Additionally, the p-value of the Cochran Q test will be reported to further evaluate the statistical heterogeneity.

**Subgroup analyses.** Subgroup analyzes will examine potential sources of heterogeneity, focusing on:

1. **Study Design:** The impact of different study designs on treatment effect estimates.

2. **Age:** Variations in prognosis and treatment response in DLBCL patients according to age.

3. **Gender:** Influence of sex on the efficacy and toxicity of chemotherapeutic regimens.

In cases of substantial heterogeneity ($I^2 > 50\%$), additional post-hoc subgroup analyses will be considered based on factors such as disease stage or specific intervention characteristics (e.g., dose, duration, route of administration).

**Meta-regression analyses.** Meta-regression will examine the associations between study-level covariates and effect size estimates using a random-effects model with restricted maximum likelihood estimation. The significance level for meta-regression will be adjusted based on the number of studies:

- **Fewer than 10 studies:** p-value $< 0.15$

- **10–20 studies:** p-value $< 0.10$

- **More than 20 studies:** p-value $< 0.05$

**Publication bias assessment.** Potential publication bias will be assessed using funnel plots and statistical tests:

- **Funnel Plots:** Visual inspection of plot symmetry.

- **Begg's** [26] **and Egger's** [27] **tests:** A p-value $< 0.1$ will indicate significant publication bias.

The trim-and-fill method, a type of sensitivity analysis, [28] will estimate the number of missing studies and adjusts the pooled effect estimate accordingly.

**Sensitivity analyses.** Sensitivity analyzes will evaluate the robustness of the overall results:

- **Leave-One-Out Method:** One study will be excluded at a time to assess its impact on the overall results [29].

- **Model comparison:** Various statistical models, weighting methods, effect measures, and heterogeneity thresholds will be employed to observe any changes in results.

**Quality analysis.** The quality of the included studies will be assessed, and their relationships with effect estimates will be examined:

- **Quality Categories:** Studies will be categorized as high, moderate, or low quality based on risk of bias.

- **Comparative Analysis:** Effect estimates of different quality categories will be compared, with the p-value and $I^2$ statistic reported for each quality analysis.

**Software and data handling.** STATA software version 14.2 software will be used to perform the meta-analysis and meta-regression analyses, manage data extraction, and assess quality. Any missing data or outliers in the primary studies will be documented along with the methods used to process them.

**Interpretation and limitations.** The results will be interpreted taking into account the size and direction of the pooled effect estimates, the precision and width of the confidence intervals, the statistical significance (p-values), the clinical significance of effect sizes, and heterogeneity measures (the $I^2$ statistic and Cochran's Q test). Implications for clinical practice and policy making, gaps in the evidence, and directions for future research will be discussed. Limitations and potential biases, such as: selective reporting, quality of primary studies, and generalizability of findings, will be acknowledged.

This protocol is intended to provide a rigorous framework for evaluating the efficacy of Pola-BR in the treatment of R/R DLBCL, thereby contributing valuable insights to clinicians and researchers in the field.

## Assessment of evidence certainty using the GRADE approach

**Approach overview.** The certainty of evidence for each outcome will be evaluated employing the Grading of Recommendations Assessment, Development, and Evaluation (GRADE) approach [30]. This method assesses the certainty of evidence based on multiple factors, including study design, risk of bias, inconsistency, indirectness, imprecision, and other relevant considerations.

## Factors assessing each outcome

1. **Risk of bias:** The methodological quality and potential for bias within the included studies will be assessed using the RoB-1 tool, as described in the specific "Risk of bias assessment" section of the protocol.

2. **Inconsistency:** Potential differences in populations, interventions, comparators and outcomes between the included studies and the review question will be examined to assess the applicability of the evidence.

3. **Indirectness:** The relevance of the evidence to the review question will be evaluated, considering variations in populations, interventions, comparators, and outcomes between studies.

4. **Imprecision:** The precision of the effect estimates will be assessed based on the confidence interval width and sample sizes of the included studies. Wider confidence intervals or smaller sample sizes may reduce the certainty of evidence.

5. **Publication bias:** A combination of the methods outlined in the Publication Bias section of the protocol will be used to assess potential publication bias.

6. **Additional considerations:** Additional factors such as large effect sizes, dose-response gradients, or potential biases will be considered to further determine the certainty of the evidence.

**Certainty ratings.** After assessing these factors, the certainty of evidence for each outcome will be categorized as "high," "moderate," "low," or "very low." A summary of results will be constructed, presenting these certainty ratings along with the effect estimates and confidence intervals for each outcome. This comprehensive approach ensures a systematic and transparent assessment of the certainty of evidence, and facilitates informed decision-making and interpretation of study results.

**Protocol amendment procedure.** If amendments to the protocol are necessary, we will document and report those amendments as follows:

1. Circumstances for amendments: Potential amendments may arise due to changes in the research question, the availability of new evidence, or methodological considerations identified during the review process.

2. Documentation: Any amendments will be documented in a log, including a description of the amendment, the rationale, the reason and the date of the change.

3. Reporting: Amendments will be reported in the final manuscript of the systematic review clearly distinguishing between pre-planned and post hoc analyses.

By adhering to these guidelines, we aim to ensure transparency and reproducibility in our systematic review and meta-analysis processes.

## Qualitative synthesis

When a meta-analysis is not possible due to insufficient data, significant heterogeneity, or other limitations, a qualitative synthesis can provide a comprehensive narrative summary of the results of the included studies. This approach systematically reviews and integrates available evidence to draw meaningful conclusions regarding the effectiveness and safety of the intervention.

**Study characteristics.**   The included studies differ in design, population, interventions and measured outcomes. These studies generally include a series of RCTs to evaluate the efficacy and safety of combination therapy with Pola-BR for the treatment of R/R DLBCL. Key characteristics such as sample size, study duration, follow-up period, and specific inclusion and exclusion criteria are described in detail to understand the context and applicability of the results of each study.

**Treatment efficacy.**   The qualitative synthesis highlights the OS and PFS outcomes reported in the studies. While some studies demonstrate a significant improvement in OS and PFS with Pola-BR compared to other treatment regimens, others indicate variability based on patient demographics, disease stage, and previous treatments. The narrative integrates these findings to provide a holistic view of the performance of Pola-BR in different subgroups of patients.

**Safety and adverse events.**   Safety profiles and AEs associated with Pola-BR are critically examined. The severity and frequency of AEs will be compared across studies to identify consistent patterns or significant concerns. The qualitative synthesis also discusses how these AEs impact the overall tolerability of the therapy.

**Heterogeneity and contextual factors.**   The heterogeneity of the study results will be examined through a detailed examination of potential sources, such as differences in study design, patient populations, interventions, and outcome measures. The synthesis examines how these factors contribute to different efficacy and safety outcomes. Contextual factors such as the setting of the studies (e.g., geographical location, healthcare system) are also considered to understand the broader applicability of the findings.

**Overall interpretation.**   The qualitative synthesis integrates findings from individual studies to provide a comprehensive account of the use of Pola-BR in the treatment of R/R DLBCL. It identifies trends, common results and outliers and provides a nuanced interpretation that takes into account the complexity and variability of the data. This approach ensures that, even without meta-analysis, meaningful and actionable insights are gained that inform clinical practice and future research.

## Discussion

R/R DLBCL represents a challenging clinical situation with limited therapeutic options and poor prognosis [31]. Although standard first-line treatment with rituximab-based

chemoimmunotherapy achieves favorable outcomes, there remains a need for effective salvation therapies for R/R DLBCL remains. This proposed systematic review and meta-analysis aim to comprehensively evaluate the effectiveness of Pola-BR as a salvage therapy for R/R DLBCL, and providing critical evidence for clinical decision-making and address the unmet need in this area.

This protocol's adherence to accepted methodological frameworks for systematic reviews and meta-analyses strengthens its design. This methodological rigor promotes a comprehensive and reliable assessment of the existing evidence. The comprehensive search strategy, which includes multiple databases and literature sources without language restrictions, increases the likelihood of capturing as many relevant studies as possible. Furthermore, the inclusion of only randomized trials and the use of the ROB-1 risk of bias assessment tool contribute to the robustness of the evidence synthesis.

However, the expected lack of appropriate studies examining the use of the Pola-BR for R/R DLBCL may limit the ability to conduct subgroup analyzes and effectively examine heterogeneity. Additionally, the novelty of Pola-BR as a treatment option may lead to the need for more available data on long-term outcomes and safety.

This systematic review will provide a comprehensive and rigorous assessment of the effectiveness of Pola-BR for R/R DLBCL, thereby filling a critical knowledge gap in this challenging area. The results will inform clinical decision-making and guideline development, and ultimately contribute to better patient outcomes. Furthermore, by identifying research gaps, this study serves as a guide for future investigations of alternative therapeutic approaches or combinations for this patient group.

## Conclusion

The aim of this systematic review protocol is to summarize the available evidence on the efficacy of Pola-BR in R/R DLBCL salvage therapy, thereby addressing an unmet clinical need. By employing a comprehensive search strategy without language restrictions, this protocol maximizes the potential to capture relevant data and provide robust evaluation. The results of this systematic review will contribute to evidence-based decision making and inform future research directions, ultimately benefiting patients with R/R DLBCL.

## Supporting information

**S1 Table. Completed checklist "Preferred Reporting Items for Systematic Review and Meta-Analysis Protocols (PRISMA-P) 2015".**
(DOCX)

## Acknowledgments

The authors would like to express their sincere gratitude to Dr. Abbas Keshtkar and the Amarafzar (Research Ware) Institute for providing invaluable training and guidance on conducting systematic reviews and meta-analyses. Their expertise and insights contributed significantly to ensuring the methodological rigor and quality of this study.

In addition, the authors acknowledge the contributions of the ChatGPT and Claude language models, which were used to improve the clarity, coherence, and overall presentation of the manuscript. These advanced language technologies facilitated the editing process, and allowed authors to iteratively to refine and improve the text iteratively.

## Author Contributions

**Conceptualization:** Mohammadreza Eslami, Mahdi Mehrabi, Fakhredin Saba.

**Methodology:** Mohammadreza Eslami, Mahdi Mehrabi, Mehrdad Payandeh, Fakhredin Saba.

**Supervision:** Mehrdad Payandeh, Fakhredin Saba.

**Writing – original draft:** Mohammadreza Eslami.

**Writing – review & editing:** Mahdi Mehrabi, Fakhredin Saba.

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
