## [Decision Letter · Decision Letter 0]

2 Jul 2024

PONE-D-24-21423Polatuzumab vedotin combined with Bendamustine and rituximab for the Relapsed/Refractory Diffuse Large B-cell Lymphoma: a systematic review protocolPLOS ONE

Dear Dr. Eslami,

Thank you for submitting your manuscript to PLOS ONE. After careful consideration, we feel that it has merit but does not fully meet PLOS ONE’s publication criteria as it currently stands. Therefore, we invite you to submit a revised version of the manuscript that addresses the points raised during the review process.

We look forward to receiving your revised manuscript.

Kind regards,

Chen Li, Ph.D.

Academic Editor

PLOS ONE

Journal Requirements:

Reviewers' comments:

Reviewer's Responses to Questions

**Comments to the Author**

1. Does the manuscript provide a valid rationale for the proposed study, with clearly identified and justified research questions?

Reviewer #1: Yes

Reviewer #2: Partly

Reviewer #3: Yes

2. Is the protocol technically sound and planned in a manner that will lead to a meaningful outcome and allow testing the stated hypotheses?

Reviewer #1: Partly

Reviewer #2: Yes

Reviewer #3: Yes

3. Is the methodology feasible and described in sufficient detail to allow the work to be replicable?

Reviewer #1: Yes

Reviewer #2: Yes

Reviewer #3: Yes

4. Have the authors described where all data underlying the findings will be made available when the study is complete?

Reviewer #1: Yes

Reviewer #2: Yes

Reviewer #3: Yes

5. Is the manuscript presented in an intelligible fashion and written in standard English?

Reviewer #1: Yes

Reviewer #2: Yes

Reviewer #3: Yes

6. Review Comments to the Author

You may also provide optional suggestions and comments to authors that they might find helpful in planning their study.

Reviewer #1: The background section of this manuscript effectively sets the stage by highlighting the clinical significance of diffuse large B-cell lymphoma (DLBCL) and the challenges associated with relapsed/refractory cases. The introduction of Polatuzumab vedotin combined with bendamustine and rituximab (Pola-BR) as a promising salvage therapy is well-articulated. However, the Abstract paragraph could benefit from briefly mentioning previous standard-of-care treatments and how Pola-BR compares in terms of mechanisms of action or preliminary efficacy. Additionally, providing more context on the prevalence and impact of DLBCL would strengthen the justification for this systematic review.

Please kindly find the attached document.

Reviewer #2: This paper provided a systematic review protocol to evaluate the efficacy and safety of using Pola-BR therapy for R/R DLBCL. I suggest some minor revisions to be finished.

1. For the purpose of this paper, I was wondering if you could expand a bit more on how following this protocol can improve the systematic reviews to evaluate the efficacy and safety of using Pola-BR therapy for R/R DLBCL.

2. For the meta-analysis objective, the author stated that “at least five studies are included to ensure a reliable statistical analysis”. How is this number determined? Why is this number sufficient to achieve reliable statistical analysis and meaningful conclusions?

3. For the subgroup analysis, why choose these 3 groups? Is any other factors can be included for subgroup analyses?

Reviewer #3: This manuscript by Mohammadreza Eslami, which uses a systematic review and meta-analysis protocol, provides a rigorous framework for evaluating the efficacy of Pola-BR in the treatment of R/R DLBCL. It is an interesting topic, but the implications of the work remain to be validated and tested experimentally.

Major:

1. Please reorganize the abstract method part and use clear sentences to describe which databases and the inception date.

2. The screening of articles, data extraction, and quality assessment should be independently reviewed and assessed by two reviewers.

3. I would suggest adding "Meta-analysis," "Protocol and guidelines," and "Systematic review" in the keywords.

4. Please use the three-line table format to create the table.

5. How are the ethical considerations addressed?

6. Please provide the disease severity and stage of diffuse Large B-cell Lymphoma in two tables.

Minor:

8. Line 10: The abbreviation "DLBCL" appears twice.

9. "MMAE" only appears once and does not need an abbreviation.

10. Please double-check the entire manuscript for abbreviations. The English full name should be followed by the English abbreviation. When the same phrase appears later, it can be replaced by the English abbreviation. If an abbreviation only appears once, please use the full name instead.

11. Please double-check the references to ensure all references use a consistent format.

7. PLOS authors have the option to publish the peer review history of their article (what does this mean?). If published, this will include your full peer review and any attached files.

Reviewer #1: No

Reviewer #2: No

Reviewer #3: No

---

## [Author Response · Author response to Decision Letter 0]

4 Jul 2024

Dear Editor,

We are pleased to resubmit our revised manuscript entitled “Polatuzumab vedotin combined with bendamustine and rituximab for relapsed/refractory diffuse large B-cell lymphoma: a systematic review protocol” for consideration in your journal.

We appreciate the time and effort you and the reviewers spent to provide valuable feedback on our manuscript. We have taken into account all comments and suggestions expressed during the review process. All questions were answered and the requested corrections were made. These changes are detailed in our point-by-point response to the reviewers, which was submitted along with this revised manuscript.

Responses to the reviewers of this manuscript are submitted in a separate file.

We believe that the revisions have significantly improved the quality and clarity of our manuscript. We hope you find the revised version for publication in your esteemed journal.

Thank you for your consideration. We look forward to hearing from you.

Kind regards,

Mohammadreza Eslami

---

## [Decision Letter · Decision Letter 1]

22 Jul 2024

Polatuzumab vedotin combined with Bendamustine and rituximab for Relapsed/Refractory Diffuse Large B-cell Lymphoma: a systematic review protocol

PONE-D-24-21423R1

Dear Dr. Mohammadreza Eslami,

We’re pleased to inform you that your manuscript has been judged scientifically suitable for publication and will be formally accepted for publication once it meets all outstanding technical requirements.

Kind regards,

Chen Li, Ph.D.

Academic Editor

PLOS ONE

Additional Editor Comments (optional):

Reviewers' comments:

Reviewer's Responses to Questions

**Comments to the Author**

1. Does the manuscript provide a valid rationale for the proposed study, with clearly identified and justified research questions?

Reviewer #1: Yes

Reviewer #2: Yes

Reviewer #3: Yes

2. Is the protocol technically sound and planned in a manner that will lead to a meaningful outcome and allow testing the stated hypotheses?

Reviewer #1: Yes

Reviewer #2: Yes

Reviewer #3: Yes

3. Is the methodology feasible and described in sufficient detail to allow the work to be replicable?

Reviewer #1: Yes

Reviewer #2: Yes

Reviewer #3: Yes

4. Have the authors described where all data underlying the findings will be made available when the study is complete?

Reviewer #1: Yes

Reviewer #2: Yes

Reviewer #3: Yes

5. Is the manuscript presented in an intelligible fashion and written in standard English?

Reviewer #1: Yes

Reviewer #2: Yes

Reviewer #3: Yes

6. Review Comments to the Author

You may also provide optional suggestions and comments to authors that they might find helpful in planning their study.

Reviewer #1: The author has comprehensively addressed all my questions. I recommend accepting the paper for publication.

Reviewer #2: The authors have thoroughly addressed all of my questions. I have no further questions and recommend proceeding with the acceptance process according to the journal's guidelines.

Reviewer #3: The authors have addressed all of my questions. I have no other comments. I recommend to accept this manuscript.

7. PLOS authors have the option to publish the peer review history of their article (what does this mean?). If published, this will include your full peer review and any attached files.

Reviewer #1: No

Reviewer #2: No

Reviewer #3: No

---

## [Editor Report · Acceptance letter]

25 Jul 2024

PONE-D-24-21423R1 

PLOS ONE

Dear Dr. Eslami, 

I'm pleased to inform you that your manuscript has been deemed suitable for publication in PLOS ONE. Congratulations! Your manuscript is now being handed over to our production team.

Kind regards, 

on behalf of

Dr. Chen Li 

Academic Editor

PLOS ONE